# Enabling Efficient, Reliable Real-World Reinforcement Learning with Approximate Physics-Based Models

**Tyler Westenbroek**
Oden Institute
University of Texas at Austin
westenbroekt@gmail.com

**Jacob Levy**
Aerospace Engineering
University of Texas at Austin
jake.levy@utexas.edu

**David Fridovich-Keil**
Aerospace Engineering
University of Texas at Austin
dfk@utexas.edu

**Abstract:** We focus on developing efficient and reliable policy optimization strategies for robot learning with real-world data. In recent years, policy gradient methods have emerged as a promising paradigm for training control policies in simulation. However, these approaches often remain too data inefficient or unreliable to train on real robotic hardware. In this paper we introduce a novel policy gradient-based policy optimization framework which systematically leverages a (possibly highly simplified) first-principles model and enables learning precise control policies with limited amounts of real-world data. Our approach 1) uses the derivatives of the model to produce sample-efficient estimates of the policy gradient and 2) uses the model to design a low-level tracking controller, which is embedded in the policy class. Theoretical analysis provides insight into how the presence of this feedback controller addresses overcomes key limitations of standalone policy gradient methods, while hardware experiments with a small car and quadruped demonstrate that our approach can learn precise control strategies reliably and with only minutes of real-world data. Code is available at https://github.com/CLeARoboticsLab/LearningWithSimpleModels.jl

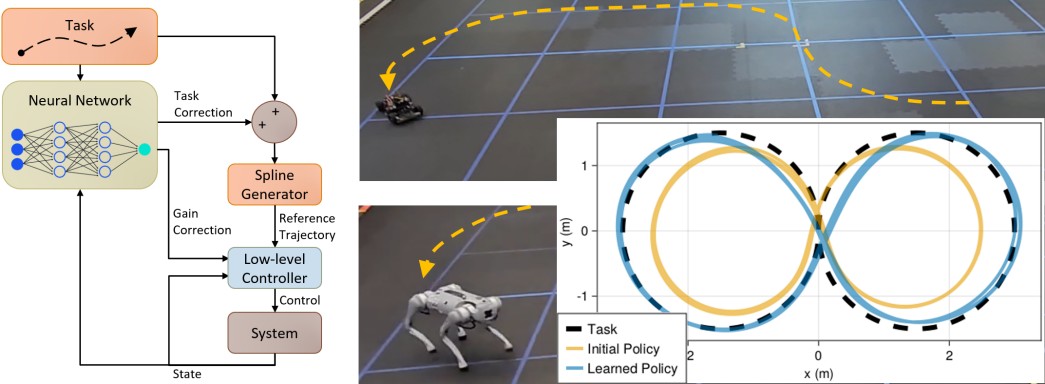

Figure 1: (Left) Schematic of the proposed policy structure, the crucial element of which is a low-level stabilizing controller which improves the smoothness properties of the underlying problem, improving learning. (Middle) Still frames depicting the approximate paths taken by a car and quadruped during test-time. (Overlaid) Top-down view of the car executing two laps of around a figure-8 before and after training.

## 1 Introduction

Reliable, high-performance robot decision making revolves around the robot's ability to learn a control policy which effectively leverages complex real-world dynamics over long time-horizons.

7th Conference on Robot Learning (CoRL 2023), Atlanta, USA.

This presents a challenge, as constructing a highly accurate physics-based model for the system using first-principles is often impractical. In recent years, reinforcement learning methods built around policy gradient estimators have emerged as a promising general paradigm for learning an effective policy using data collected from the system. However, in current practice these approaches are often too data-inefficient or unreliable to train with real hardware data, leading many approaches to train on high-fidelity simulation environments [1, 2, 3]. However, there inevitably exists a gap between simulated and physical reality, leaving room to improve policy performance in the real world. In this paper, we demonstrate how to systematically leverage a physics-based model to yield highly efficient and reliable policy optimization techniques capable of learning with real-world data.

Modern techniques for policy learning generally fall into two categories: model-free [4, 5, 6, 7] and model-based [8, 9, 10, 11, 12]. Model-free approaches learn a mapping from states to inputs directly from data. These approaches are fully general and can synthesize high-performance policies, but are extremely data-inefficient. In contrast, model-based approaches use the collected data to fit a predictive model to estimate how the system will behave at points not contained in the training set. While these approaches are more data-efficient, model inaccuracies introduce bias into policy gradient estimators [13, 14], limiting the performance of the learned policy.

However, due to the unstable nature of many robotic systems, both of these paradigms suffer from a more fundamental challenge: minute changes to the control policy can greatly impact performance over long time horizons. This "exploding gradients" phenomenon [15], [16], [17] leads the variance of policy gradient algorithms to grow exponentially with the time-horizon and renders the underlying policy learning problem ill conditioned, making gradient-based methods slow to converge [18]. Moreover, model bias also compounds rapidly over time, limiting the effectiveness of otherwise efficient model-based approaches [13].

As shown in Fig. 1, this paper systematically exploits an approximate physics-based model and low-level feedback control to overcome these challenges in policy learning. Concretely, the contributions are:

- We introduce a novel framework which uses the approximate model to simultaneously design 1) a policy gradient estimator and 2) low-level tracking controllers which we then embed into the learned policy class. Using the model to construct the gradient estimator removes the need to learn about the real-world dynamics from scratch, while the low-level feedback controller prevents gradient estimation error from "exploding".

- Theoretical analysis and illustrative examples demonstrate how we overcome exponential dependencies in the model-bias, variance and smoothness of policy gradient estimators.

- We validate our theoretical findings with a variety of simulated and physical experiments, ultimately demonstrating our method's data efficiency, run-time performance, and most importantly, ability to overcome substantial model mismatch. Overall, this paper suggests a new holistic paradigm for rapidly fine-tuning controllers using real-world data.

## 2   Related Work

Broadly speaking, there are two possible sources of bias when using a model for policy gradient estimation. The first source of error can arise if the model is used to simulate or 'hallucinate' trajectories for the system which are then added to the data set [13, 19, 20, 21]. While this approach yields a larger training set, it also introduces bias as the trajectories generated by the model can rapidly diverge from the corresponding real-world trajectory. To overcome this source of error, a number of works [14, 22, 23] have proposed policy gradient estimators which 1) collect real-world trajectories and 2) use the derivatives of a (possibly learned) model to propagate approximate policy gradient information along these trajectories. We adopt this form of estimator in this work, and note strong connections to the updates used in the Iterative Learning Control literature [24].

Evaluating the gradient along real trajectories removes the first source of error. However, inaccuracies in the derivatives of the model lead to a second source of error and, as we demonstrate in

Section 5, these errors can grow exponentially over long time horizons for unstable robotic systems. Moreover, prior works have demonstrated that exploding gradients lead to a large variance for policy gradient estimators and ill-conditioning in the underlying policy optimization problem [15], [16], [17]. We demonstrate how low-level feedback control can overcome this second source of error, while reducing variance and improving conditioning. While the use of hiearchical control architectures with embedded low-level feedback has been a key ingredient in many sim-to-real reinforcement learning frameworks [25], [26], [1], we argue that the combination of the aforementioned pieces opens the door for a new real-world training paradigm that fully leverages our approximate physics-based models.

## 3 Problem Formulation

We assume access to a simplified, physics-based model of the environment dynamics of the form:

$$x_{t+1} = \hat{F}(x_t, u_t), \tag{1}$$

where $x_t \in \mathcal{X} \subset \mathbb{R}^n$ is the *state*, $u_t \in \mathcal{U} \subset \mathbb{R}^m$ is the *input* and the (potentially nonlinear) map $\hat{F} \colon \mathcal{X} \times \mathcal{U} \to \mathcal{X}$ determines how the state evolves over discrete time steps $t \in \mathbb{N}$. To make the modelling process and down-stream controller synthesis tractable, such models are necessarily built on simplifying assumptions. For example, the model we use to control the RC car in Fig. 1 neglects physical quantities such as drag and motor time-delays. Nonetheless, such models capture the basic structure of how controller inputs will affect desired quantities (such as position) over time, and are highly useful for designing effective control architectures.

Although many reinforcement learning frameworks model the environment as a stochastic process, to aid in our analysis, we will assume that the real-world dynamics evolve deterministically, according to the (possibly nonlinear) relation:

$$x_{t+1} = F(x_t, u_t). \tag{2}$$

To control the real-world system, we will optimize over a controller architecture of the form $u_t = \pi_t^\theta(x_t)$ where $\pi^\theta = \{\pi_t^\theta\}_{t=0}^{T-1}$ represents the overall policy, $T < \infty$ is the finite horizon for the task we wish to solve, $\theta \in \Theta \subseteq \mathbb{R}^p$ is the policy parameter, and each map $\pi_t^\theta \colon \mathcal{X} \to \mathcal{U}$ is assumed to be differentiable in both $x$ and $\theta$. Thus equipped, we pose the following policy optimization problem:

$$\max_{\theta \in \Theta} \mathcal{J}(\theta) := \mathbb{E}_{x_0 \sim D}[J_T(\theta; x_0)] \quad \text{where} \quad J_T(\theta; x_0) := \sum_{t=0}^{T} R(x_t). \tag{3}$$

Here, $D$ is the probability density of the initial state $x_0$ and $R$ is the (differentiable) reward function.

## 4 Approximating the Policy Gradient with an Imprecise Dynamics Model

In this section we demonstrate how to calculate the policy gradient by differentiating the real-world dynamics map $F$ along trajectories generated by the current policy. We then introduce the estimator used in this paper, which replaces the derivatives of $F$ with the derivatives of the first-principles model $\hat{F}$. We will initially focus on the gradient $\nabla J_T(\theta; x_0)$ of the reward experienced when unrolling the policy from a single initial conditon $x_0 \in \mathcal{X}$, and then discuss how to approximate the total policy gradient $\nabla \mathcal{J}(\theta)$ using a batch estimator. To ease notation, for each $x_0 \in \mathcal{X}$ and $\theta \in \theta$ we capture the resulting real-world trajectory generated by $\pi^\theta$ via the sequence of maps defined by:

$$\phi_{t+1}^\theta(x_0) = F\big(\phi_t^\theta(x_0), \pi_t^\theta(\phi_t^\theta(x_0))\big), \quad \phi_0^\theta(x_0) = x_0.$$

**Structure of the True Policy Gradient**: Fix an initial condition $x_0 \in \mathcal{D}$ and policy parameter $\theta \in \Theta$. We let $\{x_t\}_{t=0}^T$ and $\{u_t\}_{t=0}^{T-1}$ (with $x_t = \phi_t^\theta(x_0)$ and $u_t = \pi_t^\theta(x_t)$) denote the corresponding sequences of states and inputs generated by the policy $\pi^\theta$. The policy gradient captures how changes to the controller parameters will affect the resulting trajectory and the accumulation of future rewards. The following shorthand captures the *closed-loop sensitivity* of the state and input to changes in the policy parameters: $\frac{\partial x_t}{\partial \theta} := \frac{\partial}{\partial \theta} \phi_t^\theta(x_0)$, $\frac{\partial u_t}{\partial \theta} := \frac{\partial}{\partial \theta} \pi_t^\theta(\phi_t^\theta(x_0))$ These terms depend on the derivatives of the dynamics, which we denote with:

$$A_t = \frac{\partial}{\partial x} F(x_t, u_t), \quad B_t = \frac{\partial}{\partial u} F(x_t, u_t), \quad K_t = \frac{\partial}{\partial x} \pi_t^\theta(x_t; x_0). \tag{4}$$

**Proposition 1.** *The policy gradient is given by the following expression:*

$$\nabla J_T(\theta; x_0) = \sum_{t=0}^{T} \nabla R(x_t) \cdot \frac{\partial x_t}{\partial \theta}, \ \text{where} \tag{5}$$

$$\frac{\partial x_t}{\partial \theta} = \sum_{t'=0}^{t-1} \Phi_{t,t'} B_{t'} \frac{\partial \pi_t^\theta}{\partial \theta}, \qquad \Phi_{t,t'} := \prod_{s=t'+1}^{t-1} A_t^{cl}, \qquad and \ A_t^{cl} = A_t + B_t K_t.$$

For proof of the result see the supplementary material. The first expression in (5) calculates the gradient in terms of the sensitivities $\frac{\partial x_t}{\partial \theta}$, while the latter expressions demonstrate how to compute this term using the derivatives of the model and policy. In (5) the term $\Phi_{t,t'} B_{t'}$ captures how a perturbation to the policy at time $t'$ and state $x_{t'}$ propagates through the closed-loop dynamics to affect the future state at time $t > t'$. As we investigate below, when the robotic system is unstable these terms can grow exponentially large over long time horizons, leading to the exploding gradients phenomenon and the core algorithmic challenges we seek to overcome.

**Approximating the Policy Gradient Using the Model:** We approximate the policy gradient $\nabla_\theta J_T(\theta; x_0)$ using the approximate physics-based model $\hat{F}$ in (1). Holding $x_0 \in \mathcal{X}$, $\theta \in \Theta$, and the resulting real-world trajectory $\{x_t\}_{t=0}^{T}$, $\{u_t\}_{t=0}^{T-1}$ fixed as above, we denote the derivatives of the *model* along this trajectory as:

$$\hat{A}_t = \frac{\partial}{\partial x} \hat{F}(x_t, u_t), \qquad \hat{B}_t = \frac{\partial}{\partial u} \hat{F}(x_t, u_t). \tag{6}$$

We can then construct an estimate for $\nabla J_T(\theta; x_0)$ of the form:

$$\nabla_\theta \widehat{J_T(\theta; x_0)} = \sum_{t=0}^{T} \nabla R_t(x_t) \cdot \widehat{\frac{\partial x_t}{\partial \theta}}, \ \text{where} \tag{7}$$

$$\widehat{\frac{\partial x_t}{\partial \theta}} = \sum_{t'=0}^{t-1} \hat{\Phi}_{t,t'} \hat{B}_{t'} \frac{\partial \pi_t^\theta}{\partial \theta}, \qquad \hat{\Phi}_{t,t'} := \prod_{s=t'+1}^{t-1} \hat{A}_s^{cl}, \qquad and \ \hat{A}_t^{cl} = \hat{A}_t + \hat{B}_t K_t.$$

**Remark 1.** *Note that this estimator can be evaluated by* 1) *recording the real-world trajectory which arises when policy $\pi^\theta$ is applied starting from initial state $x_0$, and then* 2) *using the derivatives of the model $\hat{F}$ to approximate the derivatives of the real-world system along that trajectory. Effectively, the only approximation here is of the form $\Phi_{t,t'} B_{t'} \approx \hat{\Phi}_{t,t'} \hat{B}_{t'}$ when calculating the estimate of the system sensitivity $\frac{\partial x_t}{\partial \theta} \approx \widehat{\frac{\partial x_t}{\partial \theta}}$. In Sections 5 and 6, we study what causes this approximation to break down over long time horizons, and how properly-structured feedback controllers can help.*

**Remark 2.** *While the policy gradient approximation given by (7) will prove convenient for analysis, this formula requires numerous 'forwards passes' to propagate derivatives forwards in time along the trajectory. As we demonstrate in the supplementary material, in practice this approximation can be computed more efficiently by 'back-propagating through time'.*

**Batch Estimation:** To approximate the gradient of the overall objective $\nabla \mathcal{J}(\theta)$, we draw $N$ initial conditions $\{x_0^i\}_{i=1}^N$ independently from the initial state distribution $D$, compute each approximate gradient $\nabla \widehat{J_T(\theta; x_0^i)}$ as in (7), and finally compute:

$$\nabla \mathcal{J}(\theta) \approx \hat{g}_T^N(\theta; \{x_0^i\}_{t=0}^T) := \frac{1}{N} \sum_{i=1}^{N} \nabla \widehat{J_T(\theta; x_0^i)}. \tag{8}$$

We use this estimator in our overall policy gradient algorithm, which is outlined in Algorithm 1.

## 5   Exploding Gradients: Key Challenges for Unstable Robotic Systems

We now dig deeper into the structure of the policy gradient and our model-based approximation. We repeatedly appeal to the following scalar linear system to illustrate how key challenges arise:

**Running Example:** Consider the case with true and modeled dynamics given respectively by:

$$x_{t+1} = F(x_t, u_t) = a x_t + b u_t \quad \text{and} \quad x_{t+1} = \hat{F}(x_t, u_t) = \hat{a} x_t + \hat{b} u_t, \tag{9}$$

---

**Algorithm 1** Policy Learning with Approximate Physical Models

---

1: **Initialize** Time horizon $T \in \mathbb{N}$, number of samples per update $N \in \mathbb{N}$, number of iterations $K \in \mathbb{N}$, step sizes $\{\alpha_k\}_{k=0}^{N-1}$ and initial policy parameters $\theta_1 \in \Theta$
2: **for** iterations $k = 1, 2, \ldots, K$ **do**
3:     **Sample** $N$ initial conditions $\{x_0^i\}_{i=1}^{N} \sim \mathcal{D}^N$
4:     **for** $i = 1, 2, \ldots, N$ **do**
5:         **Unroll** $x^i = \{\phi_t^{\theta_k}(x_0^i)\}_{t=0}^{T}$ on (2) with $\pi_t^{\theta_k}$
6:     **Estimate** $\hat{g}_T^N(\theta_k)$ using (8) and trajectories $\{x^i\}_{i=1}^{N}$
7:     **Update** $\theta_{k+1} = \theta_k + \alpha_k \hat{g}_T^N(\theta)$

---

where $a, \hat{a}, b, \hat{b} > 0$ and $x_t, u_t \in \mathbb{R}$. Suppose we optimize over policies of the form $u_t = \pi_t^\theta(x_t) = \bar{u}_t$ where $\theta = (\bar{u}_0, \bar{u}_1, \ldots, \bar{u}_{T-1}) \in \mathbb{R}^T$ are the policy parameters. In this case, the policy parameters $\{\bar{u}_t\}_{t=0}^{T-1}$ specify a sequence of open-loop control inputs applied to the system. Retaining the conventions developed above, along every choice of $\{\bar{u}_t\}_{t=0}^{T-1}$ and the resulting trajectory $\{x_t\}_{t=0}^{T}$ we have $A_t = a$, $B_t = b$, $\hat{A}_t = \hat{a}$, $\hat{B}_t = \hat{b}$ and $K_t = 0$, and thus we have $\Phi_{t,t'} = a^{t-t'-1}$ and $\hat{\Phi}_{t,t'} = \hat{a}^{t-t'-1}$. When $a, \hat{a} > 1$, the system (and model) are *passively unstable* [27, Chapter 5], and small changes to the policy compound over time, as captured by and $\|\Phi_{t,t'}\|$ and $\|\hat{\Phi}_{t,t'}\|$ growing exponentially with the difference $t - t'$, along with the formula for the gradients (5).

## 5.1 Exploding Model-Bias

Recall that the aforementioned estimator for $\nabla J_T(\theta; x_0)$ only introduces error in the term $\frac{\partial x_t}{\partial \theta} \approx \widehat{\frac{\partial x_t}{\partial \theta}}$ and in particular $\Phi_{t,t'} B_{t'} \approx \hat{\Phi}_{t,t'} \hat{B}_{t'}$ along the resulting trajectory. We will seek to understand how the point-wise errors in the derivatives of the model $\Delta A_t^{cl} := \hat{A}_t^{cl} - A_t^{cl}$ and $\Delta B_t := \hat{B}_t - B_t$ propagate over time. Towards this end we manipulate the following difference:

$$\hat{\Phi}_{t,t'} \hat{B}_{t'} - \Phi_{t,t'} B_{t'} = \Phi_{t,t'} \hat{B}_{t'} + \Delta\Phi_{t,t'} \hat{B}_{t'} - \Phi_{t,t'} B_{t'} = \Phi_{t,t'} \Delta B_{t'} + \Delta\Phi_{t,t'} \hat{B}_{t'} \quad (10)$$

$$= \Phi_{t,t'} \Delta B_{t'} + \Big( \sum_{s=t'+1}^{t-1} \Phi_{t,s} \Delta A_s^{cl} \hat{\Phi}_{s-1,t'} \Big) \hat{B}_{t'},$$

The last equality in (10) provides a clear picture of how inaccuracies in the derivatives of the model are propagated over time. For example, when approximating $\hat{\Phi}_{t,t'} \hat{B}_{t,t'} \approx \Phi_{t,t'} B_{t'}$ the error $\Delta B_{t'}$ is magnified by $\Phi_{t,t'}$, while the error $\Delta A_{t'+1}^{cl}$ is magnified by $\Phi_{t,t'+1}$.

**Running Example:** Continuing with the scalar example, in this case we have $\Delta B_t = \hat{b} - b$ and $\Delta A_t^{cl} = \hat{a} - a$. Moreover, using the preceding calculations, we have $\hat{\Phi}_{t,t'} \hat{B}_{t'} - \Phi_{t,t'} B_{t'} = a^{t-t'}(\hat{b} - b) + \sum_{s=t'+1}^{t-1} a^{t-s-1} \hat{a}^{s-t'-1} b(\hat{a} - a)$. Thus, when $a, \hat{a} > 1$ and the system is unstable, the errors in derivatives of the model are magnified exponentially over long time horizons when computing the sensitivity estimate $\frac{\partial x_t}{\partial \theta} \approx \widehat{\frac{\partial x_t}{\partial \theta}}$ and ultimately the gradient estimate $\nabla J_T(\theta; x_0) \approx \nabla \widehat{J_T(\theta; x_0)}$.

## 5.2 Exploding Variance

We next illustrate how unstable dynamics can lead our batch estimator $\hat{g}_T^N$ to explode over long time horizons $T$ unless a large number of samples $N$ are used.

**Running Example:** Consider the case where $r(x_t) = -\frac{1}{2}\|x_t\|_2^2$ and the initial state distribution is $D$ uniform over the interval $[-1, 1]$. Consider the case where we apply $\theta = (\bar{u}_1, \ldots, \bar{u}_{T-1}) = (0, \ldots, 0)$ so that no control effort is applied. In this case, for every initial condition $x_0$, the resulting state trajectory is given by $x_t = a^t x_0$, and thus our estimate for the gradient is $\nabla J_T(\theta; x_0) = \sum_{t=0}^{T-1} (a^t x_0) \cdot \sum_{t'=0}^{t-1} \hat{a}^{t-t} b$. Moreover, by inspection we see that the average of the estimator is $\mathbb{E}[\hat{g}_T^N(\theta; \{x_0\}_{i=1}^N)] = \mathbb{E}\big[ \sum_{i=1}^N \widehat{J}_T(\theta; x_0) \big] = 0$ and thus the variance of the estimator is $\frac{1}{N}\mathbb{E}[\|\hat{g}_T^N(\theta; \{x_0\}_{i=1}^N) - \mathbb{E}\big[ \sum_{i=1}^N \widehat{J}_T(\theta; x_0)\big]\|^2] = \frac{1}{N}\mathbb{E}[\|\hat{g}_T^N(\theta; \{x_0\}_{i=1}^N)\|^2] = \frac{1}{N}\|\sum_{t=0}^{T-1} (a^t x_0) \cdot \sum_{t'=0}^{t-1} \hat{a}^{t-t'} b\|^2$, a quantity which grows exponentially with the horizon $T > 0$.

## 5.3 Rapidly Fluctuating Gradients

Let $f\colon \mathbb{R}^q \to \mathbb{R}$ be a potentially non-convex and twice differentiable objective, such as the ones considered in this paper. In this general setting, well-established results for gradient-based methods characterize the rate of convergence to approximate stationary points of the underlying objective, namely, points $z \in \mathbb{R}^q$ such that $\|\nabla f(z)\|_2 < \epsilon$ for some desired tolerance $\epsilon > 0$. A key quantity which controls this rate is the smoothness of the underlying objective, which is typically characterized by assuming the existence of a constant $L > 0$ such that $\|\nabla f(z_1) - \nabla f(z_2)\| < L\|z_1 - z_2\|$ for each $z_1, z_2 \in \mathbb{R}^q$. When the constant $L$ is very large, the gradient can fluctuate rapidly, and small step-sizes may be required to maintain the stability of gradient-based methods [18], slowing the rate of convergence for these approaches. Many analyses control these fluctuations using the Hessian of the objective by setting $L := \sup_{z \in \mathbb{R}^q} \|\nabla^2 F(z)\|_{i,2}$, where $\|\cdot\|_{i,2}$ is the induced 2-norm.

Below, our main theoretical results will bound the magnitude of $\nabla^2 J(\theta)$, characterizing the smoothness of the underlying policy optimization problem and illustrating the benefits of embedded low-level controllers. We demonstrate how to derive an expression for the Hessian in the Appendix, but provide here a concrete illustration of how it can grow exponentially for unstable systems:

**Running Example:** Consider the case where the quadratic reward $r(x_t) = -\frac{1}{2}\|x_t\|_2^2$ is applied to our example scalar system. For every initial condition $x_0$ and choice of policy parameters $\theta = (\bar{u}_1, \ldots, \bar{u}_{T-1})$ by inspection we have $x_t = a^t x_0 + \sum_{s=0}^{t} a^{t-s} b \bar{u}_s$, so that the overall objective is concave and given by $J(x_0; \theta) = \sum_{t=0}^{T} \sum_{s=0}^{t-1} -\|a^t x_0 + \sum_{s=0}^{t} a^{t-s} b \bar{u}_s\|$. The Hessian of the objective can be calculated directly; in particular the diagonal entries are given by $\frac{\partial^2}{\partial \bar{u}_t^2} = \sum_{s=t+1}^{T} a^{s-t} b$. This demonstrates that $\|\nabla^2 J(x_0, \theta)\| \geq |\frac{\partial^2}{\partial \bar{u}_t^2}|_2$ grows exponentially in time horizon. From the discussion above, this implies that policy gradient methods will be very slow to converge to optimal policies.

## 6 Embedding Low-Level Feedback into the Policy Class

We now demonstrate how we can overcome the these pathologies by using the model to design stabilizing low-level feedback controllers which are then embedded into the policy class.

**Running Example:** Let us again consider the simple scalar system and model we have studied thus far, but now suppose we use the model to design a proportional tracking controller of the form $u_t = k(\bar{x}_t - x_t)$, where $\{\bar{x}_t\}_{t=0}^{T}$ represents a desired trajectory we wish to track and $k > 0$ is the feedback gain. We then embed this controller into the overall policy class by choosing the parameters to be $\theta = (\bar{x}_0, \bar{x}_1, \ldots, \bar{x}_t)$ so that $u_t = \pi_t^\theta(x_t) = k(\bar{x}_t - x_t)$. Here, the parameters of the control policy specify the desired trajectory the low-level controller is tasked with tracking. In this case, along each trajectory of the system we will now have $A_t^{cl} = a - bk$, $\hat{A}_t^{cl} = \hat{a} - \hat{b}k$, $B_t = b$ and $\hat{B}_t = b$. If the gain $k > 0$ is chosen such that $|a - bk| < 1$ and $|\hat{a} - \hat{b}k| < 1$, then the transition matrices $\hat{\Phi}_{t,t'} = (\hat{A}_t^{cl})^{t-t'-1}$ and $\Phi_{t,t'} = (A_t^{cl})^{t-t'-1}$ will both decay exponentially with the difference $t - t'$. Thus, by optimizing *through a low-level tracking controller designed with the model* we have reduced the sensitivity of trajectories to changes in the controller parameters.

**Remark 3.** *In practice, we may select a control architecture as in Fig. 1 where our parameters are those of a neural network which corrects a desired trajectory and low-level controller. The natural generalization of the damping behavior displayed by the proportional controller above is that the low-level controller is* incrementally stabilizing, *which means that for every initial condition $x_0$ and $\theta \in \Theta$ we will have $\|\Phi_{t,t'}\| \leq M\alpha^{t-t'}$. There are many systematic techniques for synthesizing incrementally stabilizing controllers using a dynamical model from the control literature [27, 28].*

We are now ready to state our main result, which demonstrates the benefits using the model to design the policy gradient estimator and embedded feedback controller:

**Theorem 1.** *Assume that 1) the first and second partial derivatives of $R_t$, $\pi_t^\theta$, $F$ and $\hat{F}$ are bounded, 2) there exists a constant $\Delta > 0$ such that for each $x_0 \in \mathcal{X}$ and $u \in \mathcal{U}$ the error in the model derivatives are bounded by $\max\{\|\frac{\partial}{\partial x}F(x,u) - \frac{\partial}{\partial x}\hat{F}(x,u)\|, \|\frac{\partial}{\partial u}F(x,u) - \frac{\partial}{\partial u}\hat{F}(x,u)\|\} < \Delta$ and*

3) *the policy class $\{\pi_t^\theta\}_{\theta \in \Theta}$ has been designed such that exists constants $M, \alpha > 0$ such that for each $x_0 \in \mathcal{X}$, $\theta \in \Theta$, and $t > t'$ we have:* $\max\{\|\Phi_{t,t'}\|, \|\hat{\Phi}_{t,t'}\|\} < M\alpha^{t'-t}$. *Letting $\bar{g}_T(\theta) = \mathbb{E}[\hat{g}_T^N(\theta; \{x_0^i\}_{i=1}^N)]$ denote the mean of our gradient estimator, there exist scalars $C, W, K > 0$ such that the bias and variance of our policy gradient estimator are bounded as follows:*

$$\|\nabla \mathcal{J}_T(\theta) - \bar{g}_T(\theta)\| \leq \begin{cases} CT^2\alpha^T\Delta & \text{if } \alpha > 1 \\ CT^2\Delta & \text{if } \alpha = 1 \\ CT\Delta & \text{if } \alpha < 1, \end{cases} \quad \mathbb{E}\left[\|\hat{g}_T^N(\theta) - \bar{g}_T(\theta)\|^2\right] \leq \begin{cases} \frac{WT^4\alpha^{2T}}{N} & \text{if } \alpha > 1 \\ \frac{WT^4}{N} & \text{if } \alpha = 1 \\ \frac{WT^2}{N} & \text{if } \alpha < 1. \end{cases}$$

*Moreover, the smoothness of the underlying policy optimization problem is characterized via:*

$$\|\nabla^2 \mathcal{J}_T(\theta)\|_2 \leq \begin{cases} KT^4\alpha^{3T} & \text{if } \alpha > 1 \\ KT^4 & \text{if } \alpha = 1 \\ KT & \text{if } \alpha < 1. \end{cases}$$

Proof of the result can be found in the supplementary material. The result formalizes the intuition built with our example: when the system is passively unstable (and we can have $\alpha > 1$), the core algorithmic challenges introduced above can arise. However, embedding a (incrementally stabilizing) low-level tracking controller into the policy class can overcome these pathologies ($\alpha \leq 1$). Note that the condition $\max\{\|\Phi_{t,t'}\|, \|\hat{\Phi}_{t,t'}\|\} < M\alpha^{t'-t}$ in the statement of Theorem 1 requires that the stabilizing controller (which has been designed for the model) is stabilizing the real-world system. This is a reasonable condition, as under mild conditions stabilizing controllers are known to be robust to moderate amounts of model uncertainty [27]. However, it is an interesting matter for future work to characterize the amount of model-mismatch our approach can handle without model-bias exploding over long time horizons.

## 7   Experimental Validation

For each experiment we use a policy structure per Fig. 1 which embeds low-level feedback that aims to stably track reference trajectories; a formal definition of this structure is given in Appendix B.1.

**NVIDIA JetRacer:** We begin by hardware-testing our approach on an NVIDIA JetRacer 1/10th scale high-speed car using the following simplified dynamics model:

$$\begin{bmatrix} x_{t+1} \\ y_{t+1} \\ v_{t+1} \\ \phi_{t+1} \end{bmatrix} = \begin{bmatrix} x_t + v_t \cos(\phi_t)\Delta t \\ y_t + v_t \sin(\phi_t)\Delta t \\ v_t + a_t\Delta t \\ \phi_t + v_t\omega_t\Delta t \end{bmatrix}, \tag{11}$$

where $\Delta t > 0$ is the discrete time-step, $(x_t, y_t, \phi_t) \in SE(2)$ are the Cartesian coordinates and heading angle of the car, $v_t > 0$ is the forward velocity of the car in its local frame, and $(a_t, \omega_t) \in \mathcal{U} = [0,1] \times [-1,1]$ are the control inputs where $a_t$ is the throttle input percentage and $\omega_t$ is the steering position of the wheels. We note that this model makes several important simplifications: (i) drag is significant on the actual car, but is missing from (11); (ii) proper scaling of the control inputs $(a_t, \omega_t)$ has been omitted; (iii) the actual car has noticeable steering bias, and does not follow a straight line when $\omega_t = 0$; and (iv) physical quantities such as time-delays in the motor are ignored.

The task consists of tracking a figure-8 made up of two circles, 3 meters in diameter, with a nominal lap time of $5.5\,\text{s}$. We implement a backstepping-based tracking controller [27, Ch. 6] for low-level control. As shown in Fig. 1 this controller alone does not ensure accurate tracking, due to inaccuracies in the model used to design it. We train a policy with $2.2\,\text{min}$ of real world data over 8 iterations, each $16.5\,\text{s}$ long and see a clear improvement in tracking performance (Fig. 1).

Next, we use a high fidelity simulator of the car to benchmark our approach against state-of-the-art reinforcement learning algorithms in Figure 2. All methods optimize over the feedback control architecture described above and therefore were trained using the same action space as our approach. We compare to the model-based approaches MBPO [13] and SVG [22] and the model-free approaches SAC [8] and PPO [9]. Each of these approaches learns about the dynamics of the system

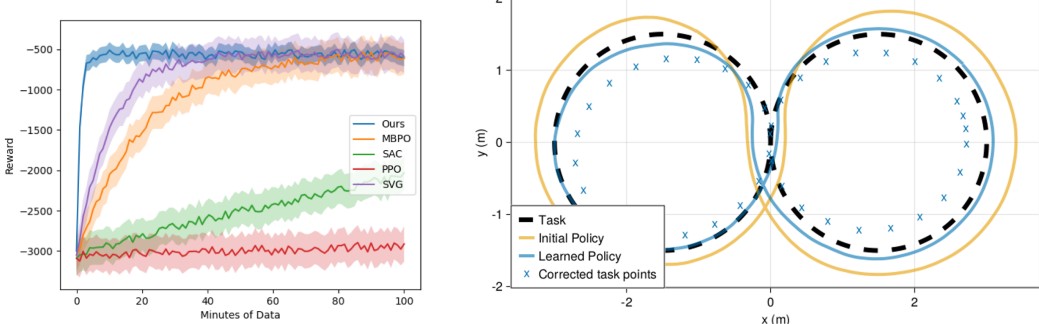

Figure 2: (Left) Training curves for different algorithms applied to a high-fidelity simulation model of an RC car. (Right) One lap of the quadruped around the figure-8 task with corrected waypoints from a neural network.

from scratch; thus, it is unsurprising that our approach converges more rapidly as it exploits known physics represented by the model. The use of feedback enables us to take this approach and obtain a high-performing controller, even though the model we use is highly inaccurate, overcoming model-bias. Additional details for the benchmark experiment can be found in Appendix B.

**Go1 Quadrupedal Robot:** We also replicate the figure-8 tracking experiment on a Unitree Go1 Edu quadrupedal robot to demonstrate the effectiveness of our approach when using a *highly simplified* model. The Go1 is an 18-degree-of-freedom system which we control in a hierarchical manner. At the lowest level, a joint control module generates individual motor torques to actuate the robot's limbs to desired angles and velocities. At the next layer, a kinematic solver converts desired foot placements to joint angles. A gait generation module determines trajectories of foot placements from high-level linear and angular velocity commands issued to the robot. We provide these high-level commands to the Go1 via Unitree's ROS interface [29], as outputs from a backstepping-based controller that was formulated using the following simplified dynamical model of the system:

$$
\begin{bmatrix} x_{t+1} \\ y_{t+1} \\ \phi_{t+1} \end{bmatrix} = \begin{bmatrix} x_t + v_t \cos(\phi_t)\,\Delta t \\ y_t + v_t \sin(\phi_t)\,\Delta t \\ \phi_t + \omega_t \Delta t \end{bmatrix}, \tag{12}
$$

where $(x_t, y_t)$ are the Cartesian coordinates of the base of the robot on the ground plane and $\phi_t$ is its heading. The two inputs to the model are the desired forward velocity $v_t$ and the desired turning rate $w_t$. Note that this is an extremely simplified model for the system, with a dynamic structure similar to the model for the car used in the previous example. Setting a nominal lap time of $37.7\,\mathrm{s}$, we trained the policy using $5.9\,\mathrm{min}$ of real-world data over 7 iterations, each $50.9\,\mathrm{s}$ long. Even though we used a highly simplified model for the dynamics, we again see a clear improvement in performance after training (cf. Fig. 2).

## 8 Limitations

Our approach successfully learns high-performance control policies using limited data acquired on physical systems. A key enabler to this end is the embedding of stabilizing low-level feedback within the policy class and the use of an *a priori* physics-based model. However, there are several key limitations. First, for situations such as contact-rich manipulation, it may not be clear how to design a controller with the required (incremental) stability property or that can incorporate necessary perceptual observations. Future work may address this limitations by incorporating techniques for learning stabilizing controllers (e.g., the Lyapunov methods of [30, 31]) or by working with latent state representations learned from vision modules. Additionally, while our method is highly sample-efficient, it does not take advantage of many established techniques from the reinforcement learning literature, such as value function learning and off policy training, leaving many directions for algorithmic advances. One particularly interesting direction is to combine the proposed approach with emerging model-based reward shaping techniques [32, 33].

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

# A   Proofs

This appendix contains proofs of claims that were omitted in the main document and several supportive Lemmas. Section A.1 provides the derivation for Proposition 1, Section A.2 states and formally derives the reverse-time representation of the gradient, while Section A.3 builds on this calculation to derive a representation for the second variation, which is subsequently use to bound the Hessian. Finally, Section A.5 contains the auxiliary lemmas.

## A.1   Proof of Proposition 1

The expression for $\nabla J_T(x_0; \theta)$ follows directly from the chain rule. To obtain the expression for $\frac{\partial x_t}{\partial \theta}$ we differentiate the dynamics $x_{t+1} = F(x_t, u_t)$ to yield:

$$\frac{\partial x_{t+1}}{\partial \theta} = \frac{\partial}{\partial x} F(x_t, u_t) \cdot \frac{\partial x_t}{\partial \theta} + \frac{\partial}{\partial u} F(x_t, u_t) \cdot \frac{\partial u_t}{\partial \theta}$$
$$= A_t^{cl} \frac{\partial x_t}{\partial \theta} + B_t \frac{\partial \pi_t^\theta}{\partial \theta},$$

where the second equality is obtained by noting that:

$$\frac{\partial u_t}{\partial \theta} = \frac{\partial \pi_t^\theta}{\partial \theta} + \frac{\partial \pi_t^\theta}{\partial x} \cdot \frac{\partial x_t}{\partial \theta} = \frac{\partial \pi_t^\theta}{\partial \theta} + K_t \cdot \frac{\partial x_t}{\partial \theta}.$$

The desired expression is then obtained by unrolling the recursion and noting that $\frac{\partial x_t}{\partial \theta} = 0$.

## A.2   Efficient Backwards Pass for Policy Gradient Computation

While the form for the policy gradient (5) and our model-based approximation in (7) will prove convenient for analysis, computing the many approximate sensitivity terms $\frac{\partial x_t}{\partial \theta}$—and in particular the $\Phi_{t,t'}$ terms—is highly complex and requires many forwards passes along the trajectory. In practice, we can more efficiently compute the approximate gradient as follows:

**Proposition 2.** *For each $x_0 \in \mathcal{X}$ and $\theta \in \Theta$ the policy gradient can be calculated via:*

$$\nabla J_T(\theta; x_0) = \sum_{t=0}^{T-1} \left( p_{t+1} B_t + \nabla R_t(x_t) \right) \cdot \frac{\partial \pi_t^\theta}{\partial \theta}, \ where \tag{13}$$

$$p_t = p_{t+1}(\hat{A}_t + \hat{B}_t K_t) + \nabla R_t(x_t) \quad and \quad p_T = \nabla R_T(x_t). \tag{14}$$

Here, the recursion with the variables $p_t \in \mathbb{R}^{1 \times n}$ performs 'back propagation through time' along the real-world trajectory using the derivatives of the model.

*Proof.* As before, let $\{x_t\}_{t=0}^{T}$ and $\{u_t\}_{t=0}^{T-1}$ denote the state trajectory that results from applying the policy $\pi_\theta$ from $x_0$.

Permitting a slight abuse of notation, we can re-write the cost by moving the dynamics constraints into the cost and weighting them with Lagrange multipliers:

$$J(\theta; x_0) = \sum_{i=0}^{T-1} R_t(x_t) + p_{t+1} \left( x_{t+1} - F(x_t, \pi_\theta^t(x_t)) \right) \tag{15}$$

Define the Hamiltonian

$$H_t(x_t, p_{t+1}, \theta) = p_{t+1} F(x_t, \pi_\theta^t(x_t)) + R_t(x_t), \tag{16}$$

and note that we may then re-write the cost as:

$$J(\theta; x_0) = R_T(x_T) + \langle p_T, x_T \rangle + \langle p_0, x_0 \rangle + \sum_{t=0}^{T-1} p_t x_t - H_t(x_t, p_{t+1}, \theta) \tag{17}$$

To reduce clutter below we will frequently omit the arguments from $H_t$, since it is clear that the map is evaluated at $(x_t, p_{t+1}, \theta)$. Let $\delta\theta \in \mathbb{R}^p$ be a variation on the policy parameters and let $\delta x_t = \frac{\partial \phi_\theta^t}{\partial \theta}\delta\theta$ denote the corresponding first variation of the state. To first order, the change in the cost corresponding to these variations is:

$$\delta J|_\theta(\delta\theta) = \langle \nabla Q_T(x_T) + p_T, \delta x_T \rangle + \sum_{t=0}^{T-1} \langle p_t - \nabla_x H_t, \delta x_t \rangle - \langle \nabla_\theta H_t, \delta\theta \rangle. \tag{18}$$

To simplify the expression, let us make the following choices for the multipliers:

$$p_T = \nabla R_T(x_T) \tag{19}$$

$$p_t^\top = \nabla_x H_t(x_t, p_{t+1}, \theta) \tag{20}$$

$$= p_{t+1}^\top \frac{\partial}{\partial x} F(x, \pi_\theta^t(x)) + \nabla R_t(x_t) \tag{21}$$

$$= p_{t+1}^\top \frac{\partial}{\partial x} A_t + \nabla R_t(x_t) \tag{22}$$

where we have applied the short-hand from developed in Section 3 for the particular task. Plugging this choice for the multipliers into (18) causes the $\delta x_t$ terms to vanish and yields:

$$\delta J|_\theta(\delta\theta) = \sum_{t=0}^{t-1} \langle \nabla_\theta H_t, \delta\theta \rangle \tag{23}$$

$$= \langle p_{t+1}^\top \frac{\partial}{\partial u} F(x, \pi_\theta^t) \frac{\partial \pi_\theta^t}{\partial \theta} + \nabla R_t(x_t) \frac{\partial \pi_\theta^t}{\partial \theta}, \delta\theta \rangle \tag{24}$$

$$= \sum_{t=0}^{T-1} \langle p_{t+1}^\top B_t + r_t, \frac{\partial \pi_\theta^t}{\partial \theta} \delta\theta \rangle \tag{25}$$

Since this calculation holds for arbitrary $\delta\theta$ this demonstrates that the gradient of the objective is given by:

$$\nabla_\theta J(\theta, x_0) = \sum_{t=0}^{T-1} \langle p_{t+1}^\top B_t + r_t, \frac{\partial \pi_\theta^t}{\partial \theta} \rangle. \tag{26}$$

$\square$

### A.3 Calculating the Second Variation

To calculate the Hessian of the objective we continue the Lagrange multiplier approach discussed above. Now let $\delta^2 x_t$ denote the second order variation in the state with respect to the perturbation $\delta\theta$. By collecting second order terms in (17) the attendant second-order variation to the cost is given by:

$$\delta^2 J|_\theta(\delta\theta) = \langle \delta x_t^\top \nabla^2 R_T(x_T), \delta x_t \rangle + \langle \nabla R_T(x_T) + p_T, \delta^2 x_T \rangle \tag{27}$$

$$+ \sum_{t=0}^{T-1} \Big( \langle p_t - \nabla_x H_t, \delta^2 x_t \rangle + \langle \delta x_t^\top \nabla_{xx}^2 H_t(x_t), \delta x_t \rangle$$

$$+ 2\langle \delta x_t \nabla_{x\theta}^2 H_t, \delta\theta \rangle + \langle \delta\theta^\top \nabla_{\theta\theta}^2 H_t, \delta\theta \rangle \Big) \tag{28}$$

By using the choice of costate introduced above, this time the second order state variations $\delta^2 x_t$ vanish from this expression so that we arrive at:

$$\delta^2 J|_\theta(\delta\theta) = + \sum_{t=0}^{T-1} \langle \delta x_t^\top \nabla_{xx}^2 H_t(x_t), \delta x_t \rangle + 2\langle \delta x_t \nabla_{x\theta}^2 H_t, \delta\theta \rangle + \langle \delta\theta^\top \nabla_{\theta\theta}^2 H_t, \delta\theta \rangle. \tag{29}$$

By unravelling this expression, we observe that:

$$\nabla^2 J_T(\theta; x_0) = \left(\frac{\partial x_T}{\partial \theta}\right)^\top \cdot \nabla^2 R_T(x_T) \cdot \frac{\partial x_T}{\partial \theta}$$

$$+ \sum_{t=0}^{T-1} \left(\frac{\partial x_t}{\partial \theta}\right)^\top \cdot \frac{\partial^2}{\partial x^2} H_t(x_t, p_t, \theta) \cdot \frac{\partial x_t}{\partial \theta}$$

$$+ 2\sum_{t=0}^{T-1} \left(\frac{\partial x_t}{\partial \theta}\right)^\top \cdot \frac{\partial^2}{\partial x \partial \theta} H_t(x_t, p_{t+1}, \theta)$$

$$+ \sum_{t=0}^{T-1} \frac{\partial^2}{\partial \theta^2} H_t(x_t, p_{t+1}, \theta),$$

which, for the purposes of our analysis, we note does not depend on second variations of the state.

## A.4  Restatement of Main Result and Proof

**Theorem 1.** Assume that the first and second partial derivatives of $R_t$, $\pi_t^\theta$, $F$ and $\hat{F}$ are bounded. Further assume that there exists a constant $\Delta > 0$ such that for each $x_0 \in \mathcal{X}$ and $u \in \mathcal{U}$ the error in the model derivatives are bounded by $\max\{\|\frac{\partial}{\partial x} F(x, u)\|, \|\frac{\partial}{\partial u} F(x, u)\|\} < \Delta$. Finally, assume that the policy class $\phi_t^\theta$ has been designed such that exists constants $M, \alpha > 0$ such that for each $x_0 \in \mathcal{X}$, $\theta \in \Theta$, and $t > t'$ we have: $\max\{\|\Phi_{t,t'}\|, \|\hat{\Phi}_{t,t'}\|\} < M\alpha^{t'-t}$. Then we may bound the bias and variance of our policy gradient estimator as follows:

$$\|\nabla \mathcal{J}_T(\theta) - \bar{g}_T(\theta)\| \le \begin{cases} CT^2 \alpha^T \Delta & \text{if } \alpha > 1 \\ CT^2 \Delta & \text{if } \alpha = 1 \\ CT\Delta & \text{if } \alpha < 1, \end{cases} \qquad \mathbb{E}\left[\|\hat{g}_T^N(\theta) - \bar{g}_T(\theta)\|^2\right] \le \begin{cases} \frac{WT^4 \alpha^{2T}}{N} & \text{if } \alpha > 1 \\ \frac{WT^4}{N} & \text{if } \alpha = 1 \\ \frac{WT^2}{N} & \text{if } \alpha < 1. \end{cases}$$

Moreover, the smoothness of the underlying policy optimization problem is characterized via:

$$\|\nabla^2 \mathcal{J}_T(\theta)\|_2 \le \begin{cases} KT^4 \alpha^{3T} & \text{if } \alpha > 1 \\ KT^4 & \text{if } \alpha = 1 \\ KT & \text{if } \alpha < 1. \end{cases}$$

*Proof.* We first bound the bias of the gradient:

$$\|\nabla \mathcal{J}_T(\theta) - \bar{g}_T(\theta)\| = \|\mathbb{E}[\nabla J_T(\theta; x_0) - \hat{g}_T(\theta; x_0)]\|$$
$$\le \mathbb{E}[\|\nabla J_T(\theta; x_0) - \hat{g}_T(\theta; x_0)\|]$$
$$\le \sup \|\nabla J_T(\theta; x_0) - \hat{g}_T(\theta; x_0)\|,$$

where the preceding expectations are over $x_0 \sim D$. The desired bound on the bias directly follows by applying the bound on gradient errors from Lemma 2 below.

Next, to bound the variance estimate note that:

$$\mathbb{E}[\|\hat{g}_T^N(\theta) - \bar{g}_T(\theta)\|^2] = \frac{1}{N^2} \sum_{i=1}^N \mathbb{E}[\|\hat{g}_T(\theta; x_0) - \bar{g}_T(\theta)\|^2]$$
$$\le \frac{1}{N} \sup \|\hat{g}_T(\theta; x_0) - \bar{g}_T(\theta)\|^2$$
$$\le \frac{4}{N} \sup \|\hat{g}_T(\theta; x_0)\|^2,$$

where the first expectation is over $(x_0^i)_{i=1}^N \sim \mathcal{D}^N$, the second is with respect $(x_0) \sim \mathcal{D}$. The desired bound on the variance follows via a direct application of Lemma 1 which provides a uniform upper-bound on the gradient estimates.

Similar to before we have:

$$\|\nabla^2 \mathcal{J}_T(\theta)\| \le \mathbb{E}_{(x_0) \sim \mathcal{D}}[\|\nabla^2 J_T(\theta; x_0)\|]$$
$$\le \sup_{(x_0) \in D} \|\nabla^2 J_T(\theta; x_0)\|.$$

The desired bound follows from Lemma 3, which uniformly bounds the task-specific Hessians. $\square$

## A.5 Supportive Lemmas

**Lemma 1.** *Let the Assumptions of Theorem 1 hold. Then there exists $\beta > 0$ independent of the parameters $T \in \mathbb{N}$, $M$ and $\alpha \in \mathbb{R}$ such that for each $x_0 \in D$ and $\theta \in \Theta$ we have:*

$$\|\nabla_\theta J_T(\theta; x_0)\| \leq \begin{cases} \beta T^2 \alpha^T & \text{if } \alpha > 1 \\ \beta T^2 & \text{if } \alpha = 1 \\ \beta T & \text{if } \alpha < 1. \end{cases}$$

*Proof.* Let the constant $L > 0$ be large enough so that it upper-bounds the norm of the first and second partial derivatives of $R_t, \pi_t^\theta$, $F$ and $\hat{F}$. Fix a specific task $x_0$ and set of policy parameters $\theta$ and let $A_t, B_t, K_t$ be defined along the corresponding trajectory as usual.

Recall from Section 3 that

$$\nabla J_T(\theta; x_0) = \sum_{t=0}^{T-1} \left(p_{t+1} B_t + \nabla R(x_t)\right) \cdot \frac{\partial \pi_t^\theta}{\partial \theta},$$

where the *co-state* $p_t \in \mathbb{R}^{1 \times n}$ is given by:

$$p_t = \sum_{s=t+1}^{T-1} \nabla R(x_t) \cdot \Phi_{s,t},$$

by inspection. Thus, we may upper-bound the growth of the co-state as follows:

$$\|p_t\| \leq LM\alpha^{T-t} + \sum_{s=t+1}^{T-1} (L + L^2)M\alpha^{s-t}. \tag{30}$$

By carrying out the summation, we observe that there exists $C_1 > 0$ sufficiently large such that

$$\|p_t\| \leq \begin{cases} C_1 T \alpha^T & \text{if } \alpha > 1 \\ C_1 T & \text{if } \alpha = 1 \\ C_1 & \text{if } \alpha < 1, \end{cases} \tag{31}$$

where we have used the fact that $\sum_{s=t+1}^{T-1} M\alpha^{s-t} < M\frac{1}{1-\alpha}$ for the third case. We can bound the overall gradient as follows:

$$\|\nabla J_T(\theta; x_0)\| = \sum_{t=0}^{T-1} L\left(L\|p_{t+1}\| + L\right), \tag{32}$$

which when combined with the bound on the costate above demonstrates the desired result for some constant $\beta > 0$ sufficiently large to cover all choices of $x_0$. $\square$

**Lemma 2.** *Let the Assumptions of Theorem 1 hold. Then there exists $C > 0$ independent of $T \in \mathbb{N}$, $M, \Delta_A, \Delta_B > 0$ and $\alpha > 0$ such that for each $x_0 \in D$ and $\theta \in \Theta$ we have:*

$$\|\nabla_\theta J_T(\theta; x_0) - \hat{g}_T(\theta; x_0)\| \leq \begin{cases} CT^3 \alpha^T \Delta & \text{if } \alpha > 1 \\ CT^3 \Delta & \text{if } \alpha = 1 \\ CT^2 \Delta & \text{if } \alpha < 1, \end{cases}$$

*where $\Delta = \min\{\Delta_A, \Delta_B\}$.*

*Proof.* Let the constant $L > 0$ be large enough so that it upper-bounds the norm of the first and second partial derivatives of $R_t, \pi_t^\theta$, $F$ and $\hat{F}$. Fix a specific task $x_0$ and set of policy parameters $\theta$ and let $A_t, B_t, K_t$ as usual.

Using equations (7) and (10) we obtain:

$$\|\nabla J_T(\theta; x_0) - \hat{g}_T(\theta, x_0)\| = \|\sum_{t=1}^{T} \nabla R(x_t) \cdot \sum_{t'=0}^{t} (\Phi_{t,t'} B_{t'} - \hat{\Phi}_{t,t'} \hat{B}_{t'})\|$$

$$\leq \sum_{t=1}^{T} \|\nabla R(x_t)\| \cdot \sum_{t'=0}^{t} \|\Phi_{t,t'} \Delta B_{t'} + \Big(\sum_{s=t'+1}^{t-1} \Phi_{t,s} \Delta A_s^{cl} \hat{\Phi}_{s-1,t'}\Big) \hat{B}_{t'}\|$$

$$\leq \sum_{t=1}^{T} L \sum_{t'=0}^{t} \Big(M\alpha^{t-t'}\Delta + \Big(\sum_{s=t'+1}^{t-1} M\alpha^{t-s}\Delta M\alpha^{s-t'}\Big)L\Big).$$

Note that the preceding analysis holds for any choice of $\theta$ and $x_0$. Thus, noting that

$$\sum_{s=t'+1}^{t-1} M\alpha^{t-s}\Delta M\alpha^{s-t'} < M^2 \frac{1}{1-\alpha}\Delta$$

in the case where $\alpha < 1$, leveraging the preceding inequality we can easily conclude that there exists $C > 0$ sufficiently large such that for each $\theta$ and $x_0$ we have:

$$\|\nabla_\theta J_T(\theta; x_0) - \hat{g}_T(\theta; x_0)\| \leq \begin{cases} CT^3\alpha^T\Delta & \text{if } \alpha > 1 \\ CT^3\Delta & \text{if } \alpha = 1 \\ CT^2\Delta & \text{if } \alpha < 1, \end{cases}$$

which demonstrates the desired result. $\qquad\square$

**Lemma 3.** *Let the Assumptions of Theorem 1 hold. Then there exists $K > 0$ independent of $T \in \mathbb{N}$, $M$ and $\alpha \in \mathbb{R}$ such that for each $x_0 \in D$ and $\theta \in \Theta$ we have:*

$$\|\nabla_\theta^2 J_T(\theta; x_0)\| \leq \begin{cases} KT^4\alpha^{3T} & \text{if } \alpha > 1 \\ KT^4 & \text{if } \alpha = 0 \\ KT & \text{if } \alpha < 1. \end{cases}$$

*Proof.* Let the constant $L > 0$ be large enough so that it upper-bounds the norm of the first and second partial derivatives of $R_t$, $\pi_t^\theta$, $F$ and $\hat{F}$. Fix a specific $x_0$ and set of policy parameters $\theta$. Recall from that the Hessian can be calculated as follows:

$$\nabla^2 J_T(\theta; x_0) = \Big(\frac{\partial x_T}{\partial \theta}\Big)^\top \cdot \nabla^2 R_T(x_T) \cdot \frac{\partial x_T}{\partial \theta}$$

$$+ \sum_{t=0}^{T-1} \Big(\frac{\partial x_t}{\partial \theta}\Big)^\top \cdot \frac{\partial^2}{\partial x^2} H_t(x_t, p_t, \theta) \cdot \frac{\partial x_t}{\partial \theta}$$

$$+ 2\sum_{t=0}^{T-1} \Big(\frac{\partial x_t}{\partial \theta}\Big)^\top \cdot \frac{\partial^2}{\partial x\partial \theta} H_t(x_t, p_{t+1}, \theta)$$

$$+ \sum_{t=0}^{T-1} \frac{\partial^2}{\partial \theta^2} H_t(x_t, p_{t+1}, \theta).$$

Using the assumptions of the theorem, we observe that there exists a constant $C_1 > 0$ sufficiently large such that

$$\max\{\frac{\partial^2}{\partial x^2} H_t(x_t, p_t, \theta), \frac{\partial^2}{\partial x\partial \theta} H_t(x_t, p_{t+1}, \theta), \frac{\partial^2}{\partial x\partial \theta} H_t(x_t, p_{t+1}, \theta)\} \leq C_1(\|p_{t+1}\| + 1) \quad (33)$$

and

$$\|\nabla^2 J_T(\theta; x_0)\| = L\|\frac{\partial x_T}{\partial \theta}\|^2 + \sum_{t=0}^{T-1} C_1(\|p_{t+1}\| + 1)\Big[\|\frac{\partial x_T}{\partial \theta}\|^2 + \|\frac{\partial x_t}{\partial \theta}\| + 1\Big] \quad (34)$$

holds for all choices of $x_0$ and $\theta$.

Using our preceding analysis, we can bound the derivative as the state trajectory as follows:

$$\|\frac{\partial x_t}{\partial \theta}\| = \|\sum_{t'=0}^{t-1} \Phi_{t,t'} B_{t'} \frac{\partial \pi_t^\theta}{\partial \theta}\|$$

$$\leq \sum_{t'=0}^{t-1} L^2 M \alpha^{t-t'}$$

This demonstrates that there exists $C_2 > 0$ sufficiently large such that:

$$\|\frac{\partial x_t}{\partial \theta}\| \leq \begin{cases} C_2 T \alpha^T & \text{if } \alpha > 1 \\ C_2 T & \text{if } \alpha = 1 \\ C_2 & \text{if } \alpha < 1, \end{cases} \tag{35}$$

where in the case where $\alpha < 1$ we have used the fact that $\sum_{t'=0}^{t-1} M \alpha^{t-t'} < M\frac{1}{1-\alpha}$. Combining the previous bounds (33), (31) and (34) then demonstrates the desired result. □

# B  Additional Experiments and Details

Here, we provide additional simulation experiments and details for experiments presented in the main paper.

## B.1  Policy Structure for Experiments

Per Section 6, for each experiment, we construct our policy (Fig. 1) around a low-level controller $\mu: \mathcal{X} \times \mathcal{X} \times \mathbb{R}^k \to \mathcal{U}$, which produces control inputs $u_t = \mu(x_t, x_t^{\text{des}}, G_t)$ to stably track reference trajectories $\{x_t^{\text{des}}\}_{t=0}^T$ with controller gains $G_t \in \mathbb{R}^k$. A task $\psi: \mathbb{R} \to \mathcal{X}$ produces the reference trajectory $x_t^{\text{des}} = \psi(t)$, however tracking is poor due to mismatch in dynamic modeling when forming the controller. To this end, a neural network $NN_\theta: \mathbb{R}^p \times \mathcal{X} \to \mathbb{R}^k \times \mathbb{R}^n$ with parameters $\theta \in \Theta$ generates corrections to the controller gains and to the reference trajectory $(\Delta G_t, \Delta x_t^{\text{des}}) = (NN_\theta^1(\xi(t), x_t), NN_\theta^2(\xi(t), x_t)) = NN_\theta(\xi(t), x_t)$, where $\xi: \mathbb{R} \to \mathbb{R}^p$ encodes task objectives. Our ultimate policy class is of the form $\pi^\theta(t, x_t; \bar{G}) = \mu(x_t, \psi(t) + NN_\theta^2(\xi(t), x_t), \bar{G} + NN_\theta^1(\xi(t), x_t))$ where $\bar{G}$ is a nominal set of feedback gains. Unless otherwise specified, the neural network is a $64 \times 64$ multilayer perceptron with $\tanh(\cdot)$ activations. For each of the benchmarks in Appendices B.4 to B.6, all methods use the same low-level feedback controller (as described in their respective sections) and the policy structure as described in this section. Therefore, all methods were trained using the same action space.

## B.2  The Benefit of Low-Level Feedback

In this experiment, we compare the policy class of Fig. 1 against a policy class in which a neural network directly determines open-loop control inputs (as in Section 5, omitting a low-level stabilizing controller). We use the double pendulum model from [34], and the task requires moving the end effector to a desired location, using a reward function based on Euclidean distance. The following parameters were used for training: Random seeds: 64, Episodes: 50, Episode length: 300, Policy calls per episode: 30, Epochs: 50, Batch size: 5. **First experiment:** We provide the true dynamics to both approaches to observe the variance and conditioning, independent of model-mismatch. Training curves for the best learning rate for each approach are depicted in Fig. 3a, which supports the our main theoretical findings. **Second Experiment:** Next we feed Algorithm 1 an approximate model that contains pendulum masses and arm lengths that are 70% and 90% of the actual values, respectively. As shown in Fig. 3b, the unstable dynamics lead to significant model bias which limited the asymptotic performance of the naive controller without embedded feedback controller.

## B.3  NVIDIA JetRacer Actor Network Outputs

We now examine neural network outputs during a single execution of the figure-eight task for the NVIDIA JetRacer hardware experiment, depicted in Fig. 4. We see that the neural network issues

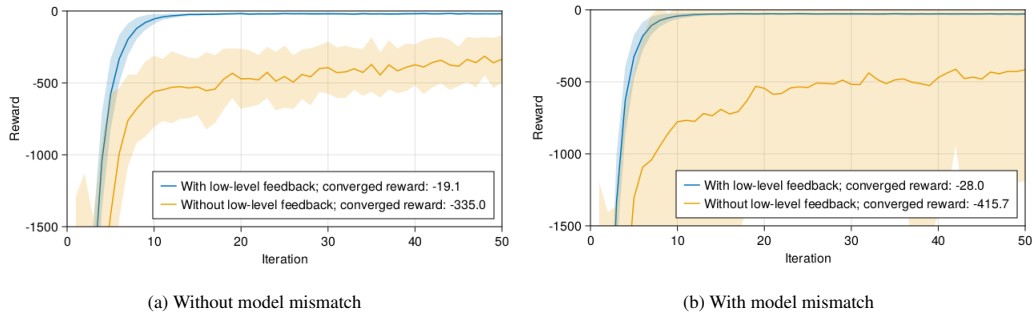

(a) Without model mismatch

(b) With model mismatch

Figure 3: Training curves for the double pendulum experiment. Embedding low-level feedback results in better performance both with and without model mismatch.

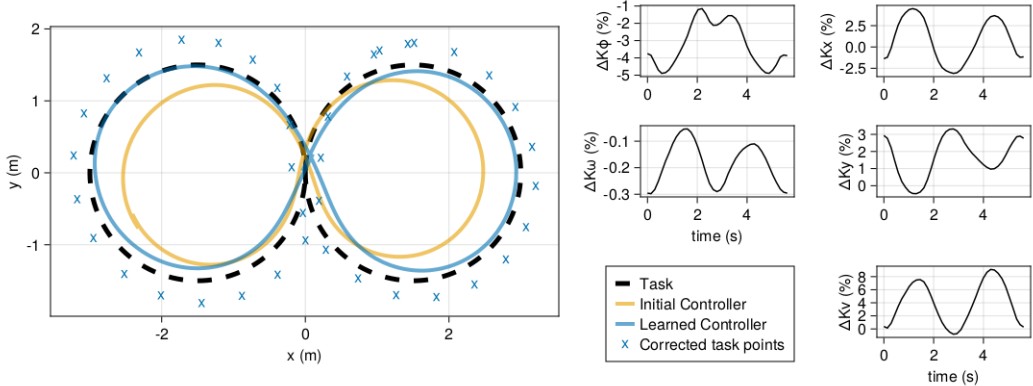

Figure 4: One lap of the car around the figure-8 task before/after training and neural network outputs.

corrections on the outside of the track, which is reasonable considering the untrained car was tracking the inside of the track. We note the following controller gains adjustments from the neural network: (i) an overall negative value selected for the feedforward steering gain $\Delta K_\omega$ counteracts the car's inherent steering bias in the positive steering direction; (ii) lower values of forward velocity gain $\Delta K_v$ were selected when crossing the origin, allowing the car to more closely track at this critical point; and (iii) elevated values of $\Delta K_v$ are selected to speed up the car for the rest of the track, increasing reward.

## B.4 Parameters for Simulated Car Benchmarks

Here, we report the details for the simulated car benchmark reported in Figure 2 in the main document. For each algorithm the episode length is 300 steps of the environment, and the simulation step was 0.1 seconds. For each method, a total of 10 random seeds was run and the actor network was a $64 \times 64$ feedforward network defining the splines tracked by the low-level controller. Further details for each tested method are:

**Our Method**: Learning rate: 0.1, Episodes per iteration: 5. **MBPO**: Dynamics model: 2 layer feedforward tanh network ($256 \times 256$), Models in ensemble: 5, Learning rate: $1 \times 10^{-3}$, Episodes per iteration: 10, Critic Network: 2 layer feedforward tanh network ($256 \times 256$). **SVG**: Dynamics model: 2 layer feedforward tanh network ($256 \times 256$), Learning rate: $2.5 \times 10^{-3}$, Episodes per iteration: 20 , Critic Network: 2 layer feedforward tanh network ($256 \times 256$). **PPO**: Episodes per iteration: 25, Learning rate: $1 \times 10^{-5}$, Critic Network: 2 layer feedforward tanh network ($256 \times 256$). **SAC:** Episodes per iteration: 5, Learning rate: $1 \times 10^{-5}$, Critic Network: 2 layer feedforward tanh network ($256 \times 256$).

### B.5 Cartpole Simulation Benchmark

In this benchmark presented in Figure 5, we attempt to track a desired end effector position for the classic simulated cart-pole environment. In particular, we use a linearizing controller [27] to approximately track desired positions for the end effector. For each algorithm the episode length was 100, and the simulation step was 0.1 seconds. For each method, a total of 10 random seeds were run and the actor network was a $64 \times 64$ feed-forward tanh network defining desired spline parameters for the desired trajectory tracked by the low-level controller.

**Our Method**: Learning rate: 0.05, Episodes per iteration: 5, Simplified Model: constructed by decreasing the mass and friction parameters of the true model by 50 percent. **MBPO**: Dynamics model: 2 layer feedforward tanh network ($256 \times 256$), Models in ensemble: 5, Learning rate: $2.5 \times 10^{-3}$, Episodes per iteration: 10, Critic Network: 2 layer feedforward tanh network ($256 \times 256$). **SVG**: Dynamics model: 2 layer feedforward tanh network ($256 \times 256$), Learning rate: $1 \times 10^{-3}$, Episodes per iteration: 20 , Critic Network: 2 layer feedforward tanh network ($256 \times 256$). **PPO:** Episodes per iteration: 25, Learning rate: $4 \times 10^{-5}$, Critic Network: 2 layer feedforward tanh network ($256 \times 256$). **SAC:** Episodes per iteration: 5, Learning rate: $5 \times 10^{-5}$, Critic Network: 2 layer feedforward tanh network ($256 \times 256$).

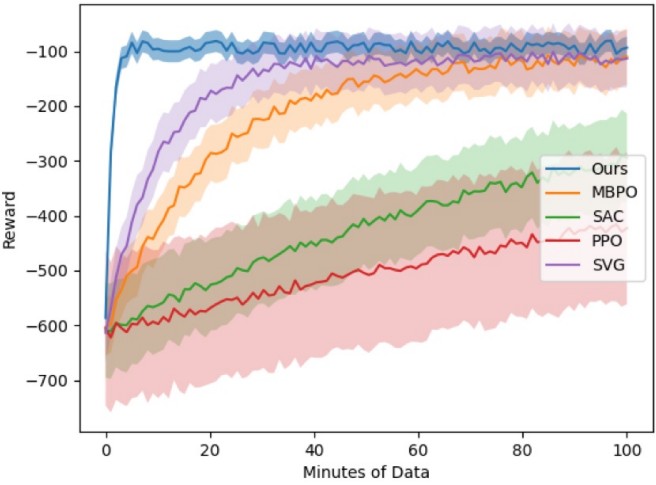

Figure 5: Training curves for different algorithms applied to the cart-pole environment.

### B.6 Quadrotor Benchmark

Next we conduct a simulation benchmark using the quadrotor dynamics model from [35] and present the results in Figure 2. The simulator timestep is 0.1s and each episode is 400 timesteps. The task is to follow a figure-8 pattern in the air. A total of 10 random seeds were run for each method, and for each algorithm the actor network was a $128 \times 128$ feed-forward tanh network, which specified the splines and feedback gains for the tracking controller from [35].

**Our Method**: Learning rate: 0.1, Episodes per iteration: 10, Simplified model: constructed by decreasing the mass and friction parameters of the true model by 50 percent. **MBPO**: Dynamics model: 2 layer feedforward tanh network ($256 \times 256$), Models in Ensemble: 5, Learning rate: $2.5 \times 10^{-4}$, Episodes per iteration: 10, Critic Network: 2 layer feedforward tanh network ($256 \times 256$). **SVG**: Dynamics model: 2 layer feedforward tanh network ($256 \times 256$), Learning rate: $1 \times 10^{-4}$, Episodes per iteration: 20 , Critic Network: 2 layer feedforward tanh network ($256 \times 256$). **PPO:** Episodes per iteration: 25, Learning rate: $1 \times 10^{-5}$, Critic Network: 2 layer feedforward tanh

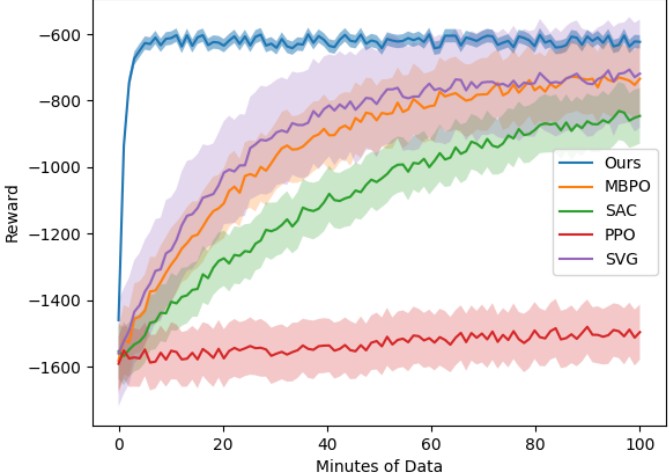

Figure 6: Training curves for different algorithms applied to the quadrotor environment.

network $(256 \times 256)$. **SAC:** Episodes per iteration: 5, Learning rate: $3 \times 10^{-5}$, Critic Network: 2 layer feedforward tanh network $(256 \times 256)$.

### B.7 Dynamics Mismatch Study – Simulated Car

In simulation for the car experiment, we study the performance of our approach as model accuracy degrades. We use the following actual dynamics model:

$$\begin{bmatrix} x_{t+1} \\ y_{t+1} \\ v_{t+1} \\ \phi_{t+1} \end{bmatrix} = \begin{bmatrix} x_t + v_t \cos{(\phi_t)} \, \Delta t \\ y_t + v_t \sin{(\phi_t)} \, \Delta t \\ v_t + \left( \beta_a a_t - c_v v_t^2 \right) \Delta t \\ \phi_t + \left( \beta_\omega \omega_t - b_\omega \right) v_t \Delta t \end{bmatrix}, \tag{36}$$

where $\beta_a$ and $\beta_\omega$ represent control input scaling for acceleration and turn rate, respectively; $c_v$ is the coefficient of drag; and $b_\omega$ represents bias in the car's steering. The set $\mathcal{A} := \{\beta_a, \beta_\omega, c_v, b_\omega\}$ parameterizes the actual dynamics of the car. We define a mismatch coefficient, $\gamma$, which scales these parameters to cause mismatch between the actual model and the model used for training. That is, we use the set $\mathcal{B} := \{\gamma\beta_a, \gamma\beta_\omega, \gamma c_v, \gamma b_\omega\}$ with Eq. (36) to form our approximate dynamics model $\hat{F}$:

$$\begin{bmatrix} x_{t+1} \\ y_{t+1} \\ v_{t+1} \\ \phi_{t+1} \end{bmatrix} = \begin{bmatrix} x_t + v_t \cos{(\phi_t)} \, \Delta t \\ y_t + v_t \sin{(\phi_t)} \, \Delta t \\ v_t + \left( \gamma\beta_a a_t - \gamma c_v v_t^2 \right) \Delta t \\ \phi_t + \left( \gamma\beta_\omega \omega_t - \gamma b_\omega \right) v_t \Delta t \end{bmatrix}. \tag{37}$$

Note that if $\gamma = 1$, then the two models match exactly.

**Training Details:** We perform training with various values of $\gamma$, over 10 random seeds, each with 15 training iterations, and present the results in Fig. 7 and Fig. 8. We find that, even in cases of large model mismatch, a policy is learned which improves the performance of the car.

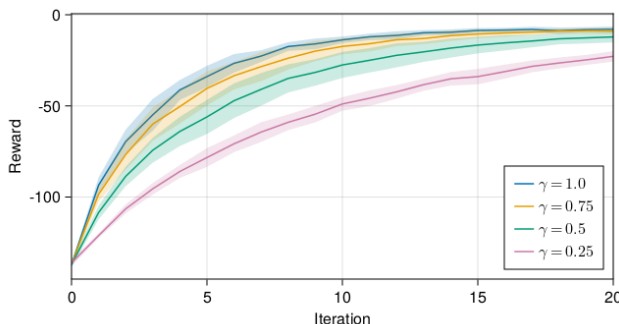

Figure 7: Training curves for the simulated car experiment for varying degrees of model mismatch.

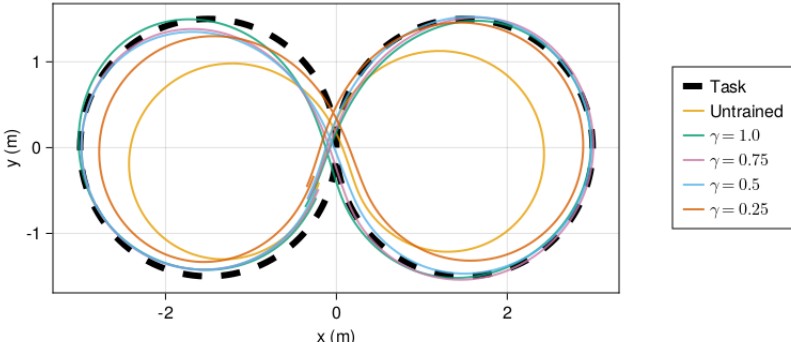

Figure 8: One lap of the car around the figure-8 task where training was performed with varying degrees of model mismatch.

