# OpenReview forum: "Enabling Efficient, Reliable Real-World Reinforcement Learning with Approximate Physics-Based Models"
_robot-learning.org/CoRL/2023/Conference — CoRL 2023 Poster_

### Official Review · Reviewer_wfR7 · 2023-07-08

**Confidence:** 3
**Originality:** Very Good
**Technical Quality:** Very Good
**Clarity Of Presentation:** Good
**Impact:** 4

**Recommendation:**

Strong Accept: I recommend accepting the paper and will argue for my recommendation even if other reviewers hold a different opinion.

**Review:**

Strengths
- The authors theoretically analyzed the reasons for the divergence of policy gradients and revealed that the proportional feedback controller can suppress the divergence.
- By making effective use of a simple model based on first principles, the proposed method achieved the policy optimization more efficiently than conventional SOTA reinforcement learning algorithms.

Weaknesses
- There are limitations in the applicability and scalability of the proposed method.
- Since a true model is required to necessarily suppress the divergence of the policy gradients, the stability should not always be guaranteed in practice.


**Quality Of The Limitations Section:**

Additional details required

**Questions For Rebuttal:**

In Fig. 2, the performance of the proposed method seems to improve when there is a model mismatch, is this an illusion due to the different vertical axes?
Anyway, it may be useful to include the convergence values in Fig. 2 to show that the final performance of the proposed method would not depend on the accuracy of the model.

I think that the model hat F that the proposed method should prepare in advance can be prepared in the same way as in conventional model-based control methods for tasks with few external factors such as this experiment.
However, what about tasks with many external factors involved, such as contact-rich robot manipulation for various objects?
In particular, in recent RL, the exact state in which the MDP is established is not known, and redundant observations (e.g. camera images) are often used.
In such cases, if the proposed method is used straightforwardly, Hessian's computational cost may be too high, and it would be very helpful to have a countermeasure to take advantage of it.


**Robotics Focus:**

Sufficient demonstration on hardware

**Summary Of Paper:**

This paper analytically clarified the destabilizing factors of the policy gradient method utilizing a simplified (imprecise) dynamics model, and showed that using a proportional feedback controller as the policy is effective for its stabilization.
The paper also demonstrated by using two types of robots that the policy optimization can be extremely efficiently achieved.

**Summary Of Recommendation:**

This is a very interesting study because it can be regarded as a method that successfully follows the conventional model-based control theory.
The usefulness of the method has been theoretically and practically verified.
However, the proposed method might not be used for arbitrary tasks with free lunch, so additional discussion would be needed on its limitations and potential for further developments.

---

### Official Review · Reviewer_D3r9 · 2023-07-10

**Confidence:** 3
**Originality:** Fair
**Technical Quality:** Very Good
**Clarity Of Presentation:** Good
**Impact:** 3

**Recommendation:**

Weak Accept: I recommend accepting the paper, but will not argue for my recommendation if the majority of other reviewers have a different opinion.

**Review:**

### Strengths
- The presentation is clear and the authors present their mathematical analysis well.
- The hardware demonstrations are compelling.

### Weaknesses
- The paper's first main conclusion, that the gradients can explode or have very high variance when the system is closed-loop unstable, has been studied before. See https://arxiv.org/abs/2202.00817 and https://arxiv.org/abs/2111.05803. Can you please clarify what your contribution is beyond these papers, either in terms of problem setting or theoretical analysis?
- The other main conclusion, that policy learning can be accelerated by embedding a lower-level feedback controller along with a deep policy, also does not seem to be new. For instance, the state-of-the-practice in RL-based locomotion seems to involve using a deep policy to predict setpoints for a stable linear joint angle tracking controller. Can you clarify your contribution beyond what is commonly done in the field today?

### Major comments
- It would seem that the main conclusion of the paper is that RL can work better when the policy is closed-loop stable. Embedding a low-level policy, as the authors suggest, is just one way of doing this. Are there other ways in which you might constrain the policy class to be stabilizing?

**Quality Of The Limitations Section:**

Limitations are addressed clearly

**Questions For Rebuttal:**

Please answer the questions raised in the previous section.

**Robotics Focus:**

Sufficient demonstration on hardware

**Summary Of Paper:**

This paper studies the behavior of gradients in the case when the forward pass is conducted on hardware and the backwards pass is conducted using some simplified approximate mode. The authors derive the error in the resulting gradient estimate and discuss factors that influence the growth of that error (particularly closed-loop stability). They show that incorporating a default-stable low-level controller beneath a high-level RL policy improves training performance.

**Summary Of Recommendation:**

The paper is well executed and discusses several critical difficulties in using analytical gradients for policy learning. However, these issues with gradients are well-known in the community, so the additional contribution of this paper is not clear. If there is specific novelty in the problem setting considered here (forward pass on HW, backwards pass on an approximate model), then I could consider upgrading my recommendation (although this still seems like an incremental contribution). In addition, the recommendation to embed a stable low-level controller is already a well-known and commonly-practiced technique in robotics.

**Update post-rebuttal**: The authors have sufficiently addressed my concerns. I still think that the main finding is not surprising, but I also think the community would benefit from the discussion in this paper, so I will increase my score.

---

### Official Review · Reviewer_wqLE · 2023-07-19

**Confidence:** 4
**Originality:** Very Good
**Technical Quality:** Very Good
**Clarity Of Presentation:** Excellent
**Impact:** 4

**Recommendation:**

Strong Accept: I recommend accepting the paper and will argue for my recommendation even if other reviewers hold a different opinion.

**Review:**

Overall, the paper is interesting and offers a good contribution. However, some points should be addressed, which would help bring confidence to the presented results.

Pros:
- Well motivated against prior work.
- Well written and clearly presented.
- The analysis is thorough, and good theoretical justification is provided.
- Solution is clear and elegant.
- Nice physical experiments on multiple platforms.

Cons:
- Unclear how novel the policy gradient estimator (Algorithm 1) is compared to prior work. Apart from the feedback controller, how does Equation (8) and Algorithm 1 differ from prior work on policy gradient estimation using dynamics derivatives ? Can you make comparisons to these other policy gradient estimators, at least in simulation?
- Providing more results in simulation using common environments (ex. Cartpole) against baselines (ex. Stochastic Value gradients, MBPO, SAC, etc.) would help solidify the results and against prior work, and help with reproducibility.
- Provide information on number of random seeds, episodes, etc. used for simulated experiments. This would help to ascertain statistical significance and confidence in results.


Comments & corrections:
- Code release would be highly suggested.
- The policy is defined with possibly time-varying parameters (line 84). In the running example used to in section 4.3, you claim the hessian of the objective is block-diagonal. Could you show this?
- There may be circumflex symbol missing above F in the second equation in 6, since the first-principles model is used here.
- Typo on line 171: should be “stochastic gradient”
- Line 184: remove redundant “the”
- line 232: remove redundant “the”

**Quality Of The Limitations Section:**

Limitations are addressed clearly

**Questions For Rebuttal:**

Please address the points listed above.

**Robotics Focus:**

Sufficient demonstration on hardware

**Summary Of Paper:**

The paper addresses the issue of model bias and variance reduction in policy gradient estimation for robot control. They provide a solution to mitigate errors and instabilities which may arise as a result. Specifically, a model-based approach for gradient estimation is presented, where real-world rollout data is used for objective/reward evaluation over a finite time horizon. This is followed by a back-propagation scheme through the dynamics to obtain the policy gradient, where the gradients of the dynamics are approximated using an analytic model. The authors examine how modelling errors and unstable dynamics affect the gradient estimate, providing ample theoretical justification. They propose to incorporate an incrementally stabilizing feedback controller to counteract these phenomena, resulting in improved policy gradient estimation. The approach is tested in simulation, against common online-RL baselines, and on multiple real-world platforms.

**Summary Of Recommendation:**

Recommended accept, on the condition that the comments are addressed (particularly better simulated experiments and more baselines.)

---

### Official Review · Reviewer_9dCJ · 2023-07-21

**Confidence:** 3
**Originality:** Good
**Technical Quality:** Good
**Clarity Of Presentation:** Very Good
**Impact:** 4

**Recommendation:**

Weak Accept: I recommend accepting the paper, but will not argue for my recommendation if the majority of other reviewers have a different opinion.

**Review:**

The paper's ideas constitute an interesting and comprehensive methodology to fuse policy-gradient/RL for continuous control methods with traditional modeling of dynamical systems and the development of model-based controllers. Although it does not address the challenges that are faced when the latter two are inevitably uncertain or impossible, it is a valuable contribution that goes beyond the mere pitching of learning-based vs non-learning-based approaches as well as "only" model-based learning approaches.
I would have liked to see also comparisons with the performance of increasing (from ideal to worst) model inaccuracy.

- Please clarify with an example how model-based approaches "inevitably introduce bias"
- Please cross-reference from the introduction the concepts of exploding gradients and exponential dependencies that are formally defined in later sections
- The first paragraph of section 7 contains repeated information and would be improved with a formal/formula definition of the NN input and output.

**Quality Of The Limitations Section:**

Limitations are addressed clearly

**Questions For Rebuttal:**

- In Figure 1, could you add the behavior of a policy with an exact model?
- Are there connections between the problems highlighted in section 5 and RL's deadly triad (e.g. model-bias and function approximation, bootstrapping/off-policy learning, and leveraging—or not—a known model/feedback controller)?
- "The policy was trained with 2.2min." Is this sentence complete?

**Robotics Focus:**

Sufficient demonstration on hardware

**Summary Of Paper:**

The paper proposes to data inefficiency and numerical instability of policy gradient reinforcement learning by using the domain knowledge of a feedback controller of (possibly low fidelity) model of the robot.
The core aspects of the methodology are (a) the use of an--inaccurate model--of the physics of the robot to help approximate the gradient of the policy model and (b) to use a low-level feedback controller (based on the inaccurate model) as a component, together with a feed-forward neural network (that computes corrected gain for the controller from the state of the system), of a learning-based control policy.
The method is demonstrated in real robots using a small car and a quadruped tracking a figure-8 trajectory.

**Summary Of Recommendation:**

The major caveats remain (1) how difficult is it to derive a model of the system and/or how inaccurate can it be for it to still yield usable results and (2) on the flip side, how much more value/challenge there is in learning a policy for a system for which we already have a feedback controller. Nonetheless, I think there is a lot of insightful observation in this paper on how to combine non-learning-based control with policy gradient reinforcement learning and the arc from theory to real robot deployment is promising.

I have read the rebuttal and appreciate the authors answer, my feedback, which is positive, is unchanged.

---

### Author Response · Authors · 2023-08-08
**Summary of Key Clarifications and Changes to Draft**

We would like to thank the reviewers for their careful reviews of the submission. We have incorporated their feedback into a new draft. Portions of the paper that are new or have undergone major revisions are highlighted with blue text. Here, we would like to summarize the revisions to the paper and our responses to reviewers on several key points.
$$
\textbf{How is the policy gradient estimator we use different from prior works which also leverage dynamics derivatives?}
$$
During the rebuttal, we have emphasized that our intention was not to claim that the class of policy gradient estimators is novel. Instead, the novelty of our approach lies in how we 1) use a simplified first-principles model to compute the estimator (removing the need for fitting a potentially complex dynamics model with real data) and 2) use the model to design and embed low-level controllers in the policy class.

We demonstrate theoretically that this overcomes exploding bias, variance, and numerical issues with this class of estimators. We demonstrate empirically that our approach $\textbf{requires an order of magnitude less data than standard approaches}$, and enables reliable, precise real-world robot learning.

$$
\textbf{How does our work compare to prior works which study exploding variance for policy gradient estimators?}
$$

While other works have pointed out the exploding variance phenomena, we are the first to explicitly demonstrate how low-level feedback control can overcome this challenge. We have added to the related work to make this point clear.
$$
\textbf{We have added the following additional evaluations to improve the submission: }
$$
---> New simulations with the car investigating how different levels of approximate model quality affect our approach. We demonstrate that the controller obtained by our approach is still highly performant, even under extreme model uncertainty.

---> New benchmark comparisons on simulated cartpole and quadrotor environments.

---> Each benchmark has been run across 10 random seeds, and we have included more information on algorithmic hyper parameters to improve reproducibility.

We are still actively working on further improving the manuscript, and will highlight any additional developments here.

---

### Decision · Program_Chairs · 2023-08-30

**Decision:**

Accept (Poster)

**Comment:**

The paper studies ways to achieve efficient learning on real systems by using an approximate dynamics model. A gradient estimator is defined that makes use of the model. Instabilities that may arise due to model bias are addressed by a feedback controller. The approach is tested in simulation and compared to common online-RL baselines. The authors also apply and evaluated it on multiple real-world platforms.

Issues raised by the reviewers were able to be cleared during the rebuttal and all reviewers recommend acceptance of the paper.

One issue that is remaining is the title:  "Feedback is all you need"
 * is not what your paper is proposing, as there are a lot of other  aspects, so it is misleading
 * the term is already overused
So I ask the authors to change the title accordingly.